# Vimentin-Positive Circulating Tumor Cells as Diagnostic and Prognostic Biomarkers in Patients with Biliary Tract Cancer

**DOI:** 10.3390/jcm10194435

**Published:** 2021-09-27

**Authors:** Sung Yong Han, Sung Hee Park, Hyun Suk Ko, Aelee Jang, Hyung Il Seo, So Jeong Lee, Gwang Ha Kim, Dong Uk Kim

**Affiliations:** 1Division of Gastroenterology and Hepatology, Department of Internal Medicine, Biomedical Research Institute, Pusan National University Hospital, Pusan National University College of Medicine, Busan 49241, Korea; mirsaint@hanmail.net (S.Y.H.); scaletlee@hanmail.net (S.H.P.); hwtkjhs@hanmail.net (H.S.K.); doc0224@pusan.ac.kr (G.H.K.); 2Department of Nursing, University of Ulsan, Ulsan 44610, Korea; jal0008@naver.com; 3Department of Surgery, Biomedical Research Institute, Pusan National University Hospital, Pusan National University College of Medicine, Busan 49241, Korea; seohi71@hanmail.net; 4Department of Pathology, Biomedical Research Institute, Pusan National University Hospital, Pusan National University College of Medicine, Busan 49241, Korea; gag86@naver.com

**Keywords:** biliary tract cancer, circulating tumor cell, vimentin, diagnosis, prognosis

## Abstract

Biliary tract cancer (BTC) has poor prognosis; thus, early diagnosis is important to decrease mortality. Although vimentin-positive circulating tumor cells (V-CTCs) are a good candidate for diagnostic and prognostic biomarkers, studies on the topic are limited. We aimed to evaluate the diagnostic efficacy of V-CTCs between BTC and benign biliary disease (BBD) and determine the prognostic value of V-CTCs in BTC patients. We recruited 69 participants who had BTCs and BBDs from a single tertiary referral center. We analyzed CTCs and V-CTCs in peripheral blood using the CD-PRIME^TM^ system. Seven patients were excluded due to a technical failure of CTC detection. CTCs were detected in all 62 patients. CTC count > 40/mL blood (55.8% vs. 20%, *p* = 0.039), V-CTC count > 15/mL blood (57.7% vs. 10%, *p* = 0.005), and V-CTC/CTC ratio > 40% (48.1% vs. 10%, *p* = 0.025) were significantly different between BTCs and BBDs. Two or more of these three parameters (61.5% vs. 10%, *p* = 0.002) increased the accuracy. A combination of CTC markers with CA19-9 and biopsy increased the accuracy (90.4% vs. 10%, *p* = 0.000). V-CTC > 50/mL blood was a significant factor affecting survival (140 (66.6–213.3) vs. 253 (163.9–342.1) days, *p* = 0.008). V-CTC could be a potential biomarker for early diagnosis and predicting prognosis in patients with BTC.

## 1. Introduction

Biliary tract cancer (BTC) is a rare type of cancer that occurs in 2–3 per 100,000 persons. The incidence is more than two times higher in northeast Asia than in other countries. However, the incidence is increasing worldwide, particularly in western countries. Furthermore, the mortality rate is relatively high compared to those of other gastrointestinal malignancies, despite the development of therapeutic agents [1,2,3,4,5]. The poor prognosis of BTC is largely due to delayed diagnosis from late examination because of non-specific symptoms such as dyspepsia, weight loss, and abdominal discomfort in the early disease stage. Additionally, BTC tissues are paucicellular with abundant fibrous stroma, leading to false negatives in pathology and resulting in late diagnosis and poor prognosis. Therefore, an exact early diagnostic method is needed for the improvement of prognosis of BTC patients.

Circulating tumor cells (CTCs) are good candidates for diagnostic or prognostic biomarkers because they enable frequent, non-invasive analysis and provide real-time dynamics of BTC. Efficient technologies for CTC analysis have been developed since the U.S. Food and Drug Administration approved the CellSearch system for clinical use to detect CTCs in peripheral blood in January 2004 [6,7,8]; however, CTC isolation and characterization remains challenging due to their rarity and heterogeneity. The use of CTCs to predict clinical outcomes is far from being applied in the real world, but these applications are being actively researched since efficient CTC enrichment is possible with recent technological advances. A centrifugal microfluidic device with fluid-assisted separation technology (FAST disc) enables label-free CTC isolation from whole blood in a size-selective manner. This system uses tangential flow filtration (TFF), which allows clog-free, ultrafast (>3 mL/min) CTC enrichment with gentle reductions in pressure (~1 kPa) for collecting a large amount of tumor cells with high viability.

CTCs are detected even in precancerous lesions by circulating along blood vessels through the epithelial to mesenchymal process [9,10]. Thus, tumor detection could be possible by detecting CTCs, especially vimentin-positive CTCs developed during the epithelial-mesenchymal transition (EMT) process in the early disease stage. Furthermore, vimentin expression in CTCs is possibly highly correlated with cancer progression rather than CTCs [11,12].

There are limited studies on using CTCs for early tumor detection and prognosis of BTC. Further, the cut-off for a positive CTC value has not yet been defined. The aim of this study was to evaluate the diagnostic efficacy of vimentin-positive circulating tumor cells (V-CTCs) in BTCs and benign biliary diseases (BBDs). Additionally, we aimed to determine the prognostic value of V-CTCs in BTC patients.

## 2. Materials and Methods

### 2.1. Patient Characteristics

We recruited 69 participants from a single tertiary referral center in South Korea between June 2018 and February 2021. The inclusion criteria for BTCs were (1) age ≥ 18 years; (2) BTC diagnosis based on ultrasound (US), computed tomography (CT), and magnetic resonance imaging (MRI); and (3) histological confirmation as adenocarcinoma. The inclusion criteria for BBDs were (1) age ≥ 18 years; (2) benign biliary diseases such as cholelithiasis and benign biliary stricture based on US, CT, and MRI; and (3) no history of other malignancies within 5 years. Blood samples were collected at the initial visit. Seven patients were excluded due to a technical failure of CTC detection. Finally, 62 patients were enrolled for the assessment of CTC number (Figure 1).

Patients were followed clinically using medical records to determine treatment regimens and responses, including surgery, disease progression, and time of death. This prospective trial was conducted at a single tertiary medical center with institutional review board approval (H-H-1801-020-062), and all patients provided written informed consent. The Clinical Research Information Service (CRIS) approved this study (KCT0003511).

### 2.2. CTC Enumeration and Characterization

For all cases, peripheral blood samples were maintained at room temperature and pretreatment was performed within 2 h of collection. We used a CD-PRIME^TM^ system (Clinomics, Ulsan, Korea) which is a commercialized version of the FAST disc. The system contains two parts, the CD-CTC^TM^ Duo (disc) and a CD-OPR-1000^TM^ (disc operating machine); we collected intact CTCs from the white buffy coat resuspended with phosphate buffered saline (PBS) in the same amount as the original blood of each BTC patient.

Immunostaining was performed to identify the isolated cells on the membrane in the filtration chamber of the FAST disc. The isolated cells were stained with fluorescence-conjugated antibodies, including FITC conjugated anti-EpCAM antibody (1:417, 9C4; BioLegend, San Diego, CA, USA), Alexa488 conjugated anti-pan-cytokeratin antibody (1:100, AE1/AE3; Invitrogen, Carlsbad, CA, USA), FITC conjugated anti-cytokeratin antibody (1:500, CAM5.2; BD Biosciences, San Diego, CA, USA), Alexa555 conjugated anti-vimentin antibody (1:125, D21H3; Cell signaling, MA, USA), and Cy5 conjugated anti-CD45 antibody (1:50, F10-89-4; Southern biotech, Birmingham, AL, USA) in PBS with 0.01% tween 20 and mounted with 4,6-diamidino-2-phenylindole (DAPI, Abcam, Cambridge, CB2, UK).

The cells were fixed in 4% paraformaldehyde for 20 min and stained with surface antibodies (CD45, EpCAM) in the dark for 20 min. Then, the cells were permeabilized with 0.01% Triton-X for 10 min and stained with intracellular antibodies (cytokeratin, pan-cytokeratin, and vimentin) in the dark for 20 min. Finally, the cells were stained with DAPI and examined under a fluorescence microscope. All staining processes were performed at room temperature, and the cells were washed with PBS in each step.

Cells were counted as CTCs if they had intact morphology (large cell with an intact nucleus; cut-off size of CTCs is 8 μm), stained positive for EpCAM, pan-cytokeratin, cytokeratin, and DAPI, and stained negative for CD45 by researchers blinded to the patient clinical status. In addition, V-CTCs were referred to positive staining for vimentin.

To validate the expression of CTCs, we spiked 100 cells of BTC cell lines such as SNU-1079, SNU-308, and SNU-1196 in 3 mL blood of healthy subjects. After the enrichment of cancer cells from spike-in blood, the cancer cells on the membrane were stained with fluorescence-conjugated antibodies and confirmed the CTCs expression marker.

### 2.3. Outcome Assessment

The primary study endpoint was to reveal the relationship between baseline CTC counts, V-CTC counts, V-CTC proportion, and the pathologic BTC diagnosis. The secondary endpoints were to find correlations between baseline V-CTC counts, progression free survival (PFS), and overall survival (OS).

The patients were followed for disease progression by imaging and laboratory testing. PFS was defined as the relapsed time from the time of pathologic diagnosis, and was assessed by peripheral blood sample collection, CT, MRI, and positron emission tomography (CA19-9) imaging. OS was defined as the time elapsed from the time of pathologic diagnosis until death.

### 2.4. Statistical Analysis

Statistical analysis was performed using IBM SPSS statistical software, version 21.0 (IBM Corp, Armonk, NY, USA). Descriptive statistics are presented as frequencies and percentages for categorical variables and as means ± standard deviations for continuous variables. Two or three-sample comparisons were performed using the Student’s t-tests and ANOVA test for normally distributed variables. Wilcoxon rank sum tests and Kruskal–Wallis tests were used for non-parametric comparisons. A two-sided *p*-value of <0.05 was used to indicate statistical significance in all analyses. Differences in OS were plotted using Kaplan–Meier survival plots and tested using log-rank tests. The optimal cut-off value for CTC counts was determined using receiver operating characteristic (ROC) curves and the area under the curve (AUC) values were calculated. To evaluate the factors affecting the prognosis, COX regression analysis was performed, with factors known as prognostic markers and CTC markers as variables.

## 3. Results

### 3.1. Patient Characteristics

A total of 62 patients were enrolled for the assessment of CTC markers. Of them, 10 were diagnosed with BBDs and 52 were diagnosed with BTC (8 with gallbladder cancer (GB), 12 with intrahepatic cholangiocarcinoma (IHCC), 21 with extrahepatic cholangiocarcinoma (EHCC), and 11 with perihilar cholangiocarcinoma (PHCC)). Table 1 shows the characteristics of patients with benign, resectable, and unresectable BTC. All epidemiologic factors except smoking status were the same between patients. The mean age of patients with BBDs (50% male) was 66.1 years and that of patients with BTCs (61.5% male) was 69.2 years. The alanine transaminase ALT (28.6 vs. 119.9%, *p* = 0.026), alkaline phosphatase (ALP) (139.9 vs. 326.7, *p* = 0.035), and total bilirubin (0.81 vs. 5.76, *p* = 0.048) levels were significantly different between the BBD and BTC groups, respectively. These factors are markers of biliary obstruction. The CEA (3.0 vs. 3.9 vs. 11.1, *p* = 0.021) and CA19-9 (16.0 vs. 434.3 vs. 1165.1, *p* = 0.040) levels were also significantly different between BBD, resectable BTC, and unresectable BTC groups.

### 3.2. CTC Counts in BTC and BBD

Figure 2 shows the results of CTC and V-CTC analysis in patients with unresectable and resectable BTC and patients with BBD. Though the CTC and V-CTC counts differed between the BTC and BBD groups, this difference was not significant, whereas the V-CTC/total CTC count ratio (VCR) showed a statistically significant difference between the groups (35.7% vs. 23.8%, respectively, *p* = 0.048). There were no statistically significant differences in CTC count, V-CTC count, and VCR between patients with resectable and unresectable BTC (Figure 3). The CTC count, V-CTC count, and VCR cut-off values, determined via ROC curve analysis, were 40/mL blood, 15/mL blood, and 40%, respectively (Appendix A.) When CTCs were analyzed using these cut-off values, significant difference across all three parameters were found between the BTC and BBD groups (CTC > 40: 55.8% vs. 20%, *p* = 0.039; V-CTC > 15: 57.7% vs. 10%, *p* = 0.005; VCR > 40%: 48.1% vs. 10%, *p* = 0.025, respectively). Analyzing any two of the three parameters in combination precipitated a more statistically significant difference between the BTC and BBD groups (61.5% vs. 10%, *p* = 0.002) than using any one parameter alone (*p* = 0.002). Notably, when patients showed two of three parameters plus biopsy results or elevated CA19-9 levels, the sensitivity and specificity of discrimination between BBD and BTC increased (90.4% vs. 10%, *p* < 0.001). (Table 2).

### 3.3. Subgroup Analysis: Benign vs. Resectable Biliary Tract Cancer

Table 3 shows the CTC counts of the BBD and resectable BTC groups. The indicators used to distinguish the BTC and BBD groups were applied between resectable BTC and BBDs. CTC count > 40/mL blood (54.5% vs. 20%, *p* = 0.002), V-CTC count > 15/mL blood (51.5% vs. 10%, *p* < 0.001), and VCR > 40% (45.5% vs. 10%, *p* = 0.000) were used to differentiate the two groups. Using two of the three parameters in combination (57.6% vs. 10%, *p* = 0.007) also yielded statistically significant results. Further, using a combination of these parameters, patient biopsy results, and elevated CA19-9 level data also increased the sensitivity and specificity of discriminating between BBDs and resectable BTC (Table 3).

### 3.4. Association of the CTC Count with Prognosis

In the prognostic analysis of patients with BTC using their neutrophil/lymphocyte ratio (NLR), CA19-9 level, CTC count, V-CTC count, and VCR data, V-CTC counts > 50/mL blood was found to be the most significant (Table 4.) This cut-off value of V-CTC count was determined by ROC curve analysis, under or over 250 days (mean OS = 257 ± 184 days, AUC = 0.615, sensitivity = 32.1%, specificity = 87.5%). There was no significant difference in the baseline characteristics between the groups according to V-CTC counts of 50/mL blood (Table 5). Other non-significantly different prognostic markers included the NLR. Figure 4 shows the Kaplan–Meier survival analysis. Patients BTC with V-CTC count > 50/mL blood showed a poorer prognosis than other patients with BTC (median survival: 140 (66.6–213.3) vs. 253 (163.9–342.1) days, *p* = 0.008). In patients with resectable BTC, the prognosis was significantly different between patients with V-CTC count >50/mL blood and V-CTC count < 50/mL blood (median survival: 167 (97.7–236.3) vs. 311 (254.8–367.2) days, *p* = 0.004). The median survival of the V-CTC count > 50 and count < 50 groups in subgroup analysis according to the location of the cancer was 170 (0–345.3) vs. 95 (0–224.6) days for IHCC (*p* = 0.076), 307 (267.8–346.2) vs. 218 (117.1–318.9) days for EHCC (*p* = 0.072), and 324 (6.8–443.1) vs. 138 days for GB cancer (*p* = 0.353), which was similar to the result obtained with the total number of patients, though the number of patients in each subgroup was too low to obtain meaningful results. The PHCC group showed similar median survival between the >50 and <50 V-CTC count groups (293 (141.1–444.9) vs. 245 days, *p* = 0.835) (Appendix A.) However, PFS was not significantly different between the groups in accordance with any CTC marker, except CA19-9 level (CA19-9 < 40 vs. >40: 284 (168.5–399.5) vs. 163 (152.5–217.4) days, *p* = 0.011, respectively) (Appendix A).

### 3.5. Technical Failure of CTC Detection in Patients with Biliary Tract Cancers

CTC detection failed in 7 of the 69 patients with BTC enrolled in this study, all of whom were in the advanced disease stage. In all seven patients, large amounts of amorphous necrotic matrices made it impossible to count the CTCs accurately. Further, all seven patients showed more frequent metastasis (71.4% vs. 23.1%, *p* = 0.007), significantly lower platelet counts (187k vs. 270k, *p* = 0.017), higher NLR (18.7 vs. 4.1, *p* < 0.001), and higher CA19-9 levels (4475 vs. 701, *p* < 0.001) than patients with detectable CTCs (Appendix A).

## 4. Discussion

The estimation of V-CTCs is a potential diagnostic approach for BTC, in addition to evaluating CA 19-9 levels, radiologic imaging, and core or forceps biopsy. Since the CTC markers have low diagnostic accuracy when used independently, we used the CTC markers in combination. Though evaluation using a combination of CTC markers improves accuracy, diagnosing BTC based on CTC markers alone is difficult. Thus, combining the results of this estimation with those obtained by traditional method, such as biopsy and CA19-9 level assessment, can facilitate accurate BTC diagnosis. Furthermore, the V-CTC count was related to OS, especially that of patients with resectable BTC. In multivariate analysis including CA19-9 levels, which is a well-known prognostic factor of BTC, the only significant prognostic factor was a V-CTC count > 50. However, additional studies are needed to support this result.

Efforts towards early diagnosis and prognosis prediction are constantly being made in cancer research. Recently, with the advent of precision medicine, interest in the use of target markers to provide personalized treatment, based on the systemic biology of cancer, has increased. However, there are no minimally invasive methods currently available to accurately diagnose early-stage cancer or predict cancer progression. Recently, studies have been conducted to analyze CTCs, circulating tumor DNA (ctDNA), and extracellular vesicles derived from tumors. Beyond aiding in early cancer diagnosis and prognosis determination, these circulating tumor markers form the basis of many key aspects of precision medicine, including determining actionable targets, monitoring treatment response and resistance, and selecting therapeutics.

There are two important steps in the assessment of CTCs. First, cell enrichment is performed using biological and physical properties. Then, protein-based techniques are used for positive CTC selection. This selection relies on the detection of specific markers by antibodies. However, the expression of epithelial markers such as EpCAM and pan-cytokeratin can be reduced during EMT, which can result in false negatives. Thus, mesenchymal markers, such as N-cadherin and vimentin, should be used [13]. The proportion of true mesenchymal phenotype of CTCs would be very limited because the EMT is a dynamic process when entering the circulation [14,15]. Another way to enrich CTCs is to distinguish CTCs based on their physical properties.

There are many challenges in the assessment of CTCs. The reproducibility of these assessments is difficult since the detected CTC subpopulations may vary across experiments. Additionally, CTCs are large and can be trapped in peripheral blood vessels. CTCs also undergo apoptosis 1–2 h after entering the bloodstream, which may result in low levels of CTCs being detected. Another challenge is the discrimination of CTCs from normal circulating cells. In a study involving patients with benign colonic disease, 11–19% of the patients had epithelial cells that were considered CTCs [16]. For this reason, CTC detection methods usually use epithelial cell adhesion molecules (EpCAMs), which may lead to an underestimation of CTC counts [17].

In this study, we used a platform comprising a centrifugal microfluidic device with a fluid-assisted separation filter membrane (FAST disc) to collect CTCs. The FAST disc enabled label-free CTC isolation from whole blood in a size-selective manner via tangential flow filtration (TFF). This system allowed a clog-free, ultrafast (>3 mL/min) CTC enrichment with gentle pressure drops (~1 kPa) for high viability. Since only gentle pressure was used, cells of various sizes were captured on the membrane, thus facilitating the counting of intact CTCs and allowing for the collection of a large number of tumor cells with high viability. Using vimentin to identify cells in the EMT process increased CTC counts [18,19]. In our study, high counts of CTCs, especially V-CTCs, were found even in patients with BBDs, indicating that the EMT process may also occur in BBDs.

A novel platform to diagnose BTC and predict its prognosis is required for several reasons. Patients with BTC usually present with non-specific symptoms, such as dyspepsia, weight loss, and abdominal discomfort in the early disease stage. A positive BTC diagnosis is usually only made in the later stages of the disease when overt symptoms, such as jaundice, are present. Imaging by US, CT, and MRI is effective for detecting masses in the biliary tract. However, due to the low incidence of the disease, these methods are not cost-effective for BTC diagnosis. The pathologic diagnosis of BTC is difficult due to various anatomical factors, such as the deep location of the liver, the superficial spread of the bile duct, and the complex blood vessel distribution around the tumor. Although liver core biopsy, forceps biopsy through endoscopic retrograde cholangiopancreatography (ERCP), and brush cytology through ERCP are currently available techniques, they are both invasive and unsuitable for obtaining a sufficiently large cell for pathologic diagnosis because BTC tissues are paucicellular within abundant stroma. However, it is not always possible to obtain tissue samples from primary or metastatic sites. Even if tissue samples are obtained at the time of initial diagnosis, it is not certain that they can be obtained at recurrence or during tumor progression. Therefore, we aimed to develop a method for detecting high-risk groups by screening images during early BTC diagnosis.

In early BTC research, CTCs were detected using CEA-nested RT-PCR in the nucleated cell fraction. The detection rate of CEA-mRNA was 47.8–52.5% (21 of 40 patients with biliary-pancreatic cancers), which was relatively lower than that reported in a recent study [20,21].

Through analysis using the CellSearch system, low counts of CTCs were found in patients with BTC. The detection rate of CTCs in 3 of 13 BTCs is 23.1% [22]. The 12-month survival rates of the patients in the CTC-positive and CTC-negative groups were 25% and 50%, respectively. In another study [23], 88 patients (17%) were positive for CTCs with more than two, which was an independent predictor of survival. Although CTC detection is rare, assessing CTC counts may be useful for predicting the mortality risk of BTC. However, in a recent study evaluating the therapeutic efficacy of cediranib, no relationship between CTC count detection and survival was found. Furthermore, the benefits of cediranib treatment could not be predicted by the combined analysis of baseline and cycle 3 CTC count [24].

A new marker was evaluated for the detection of more CTCs in patients with BTC. In nonconventional CTCs (ncCTCs) lacking epithelial and leukocyte markers, the positive identification of CTCs increased from 19% to 83% [25]. ncCTCs are also correlated with disease-specific survival. Using a novel glycosaminoglycan, SCH45, CTCs were detected in 65 patients with advanced BTC. Furthermore, SCH45-based CTC counts were correlated with the prognosis of patients with BTC receiving chemotherapy [26]. Ninety percent of patients with pancreatic biliary cancers expressed pan-cytokeratin or V-CTCs, which increased the diagnostic accuracy of pancreatic biliary cancers [27].

We excluded seven patients in whom CTCs were not detected from the analysis. In all seven patients, only large amounts of amorphous necrotic matrices were found. These patients had a high ratio of metastasis, high levels of CA19-9, and a higher NLR compared to the other patients. Since CTCs were mostly not detected in patients with advanced cancers, the non-detection of CTCs with extensive necrotic materials may indicate an advanced cancer stage.

There are several limitations to this study. First, this study was conducted with a small number of patients with BTC and BBDs. Second, the patients with BBDs presented with active inflammation. Although blood was drawn immediately after the infection was controlled, the active inflammation may have affected the detection of epithelial cells in circulation. Third, we did not obtain follow-up blood samples to assess the dynamics of CTCs during therapy.

Many researchers have worked to identify the best biomarkers for diagnosing early-stage BTC. However, this is made difficult by the anatomical and histological characteristics of BTC. The combined assessment of circulating tumor markers and CA 19-9 levels, radiological imaging, and core or forceps biopsy may be helpful in discriminating CTCs between early-stage BTC and BBDs, and in determining future prognosis in patients with resectable BTCs. Although there are still limitations to early BTC diagnosis using V-CTCs, further studies may provide a framework for realizing precision medicine by conducting liquid biopsies using CTCs in a complementary manner.

## Figures and Tables

**Figure 1 jcm-10-04435-f001:**
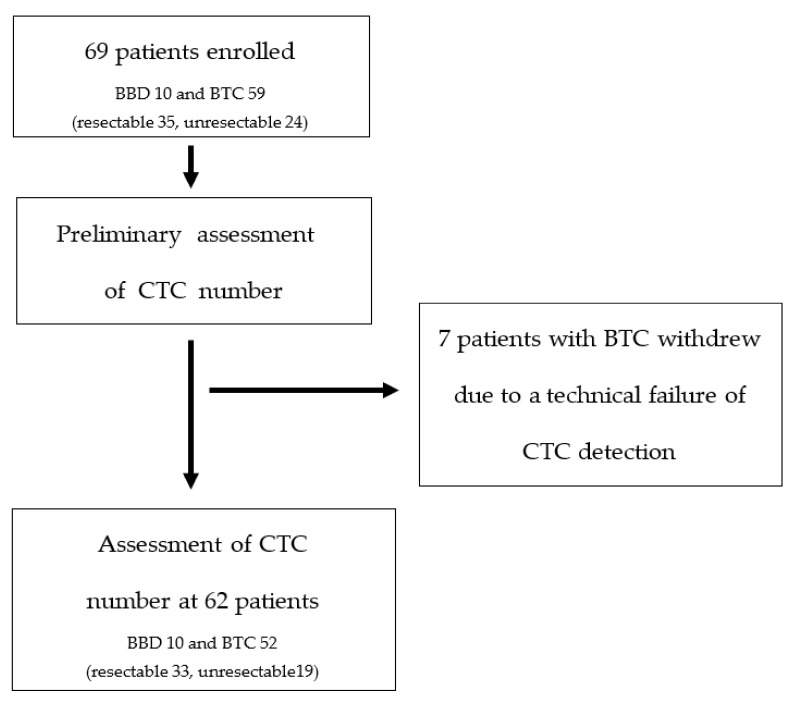
Flow chart of the study. BBD; benign biliary disease, BTC; biliary tract cancer, CTC; circulating tumor cell.

**Figure 2 jcm-10-04435-f002:**
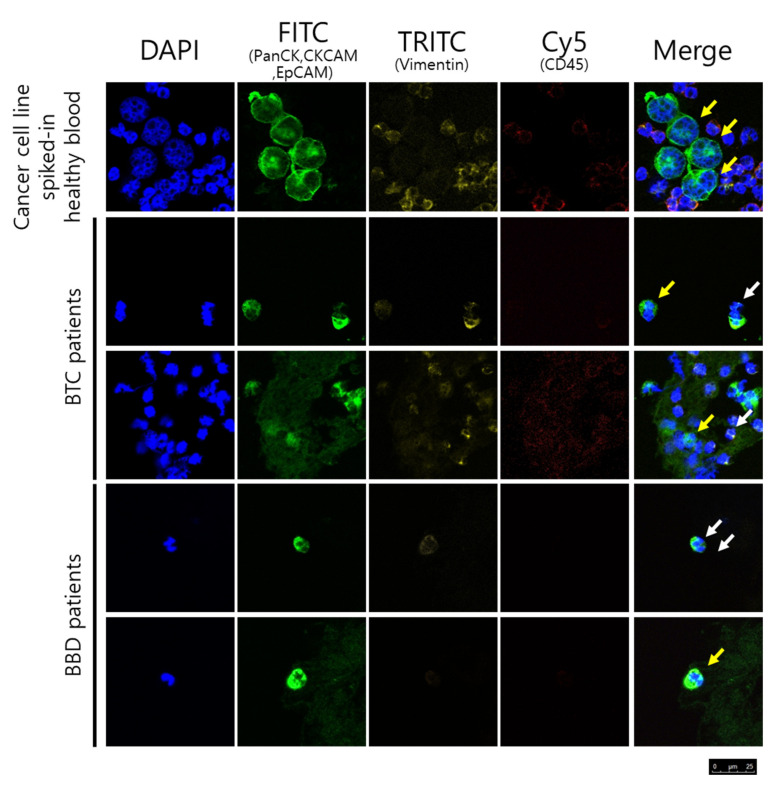
The yellow arrows indicate CTCs (PanCK+/CKCAM+/EpCAM+/CD45-) and white arrows indicate V-CTCs (PanCK+/CKCAM+/EpCAM+/CD45-, vimentin+) in BTC and BBD patients.

**Figure 3 jcm-10-04435-f003:**
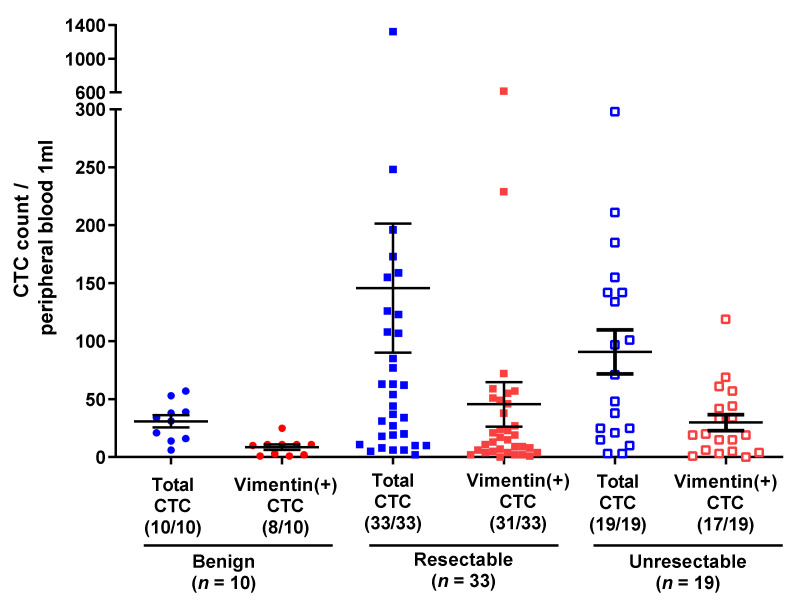
CTC counts and V-CTC counts in patients with biliary disease.

**Figure 4 jcm-10-04435-f004:**
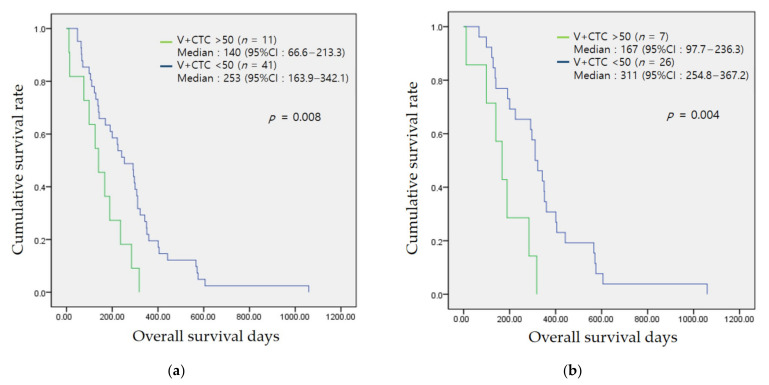
Kaplan–Meier overall survival in total (**a**), resectable (**b**) and unresectable (**c**) biliary tract cancer patients.

**Table 1 jcm-10-04435-t001:** Patient characteristics and CTC counts/v-CTC proportion.

	Benign Biliary Disease *n* = 10	Biliary Tract Cancer *n* = 52		*p*-Value
		Resectable *n* = 33	Unresectable *n* = 19	
Sex male, (%)	5 (50)	32 (61.5)		0.504
		20 (60.6)	12 (63.2)	0.536
Age	66.1 ± 8.2	69.2 ± 10.8		0.393
		71.4 ± 9.7	65.4 ± 11.8	0.095
Diagnosis	IHD stone/CBD stone/benign biliary stricture 3 (30)/3 (30)/4 (40)	GB/IHCC/EHCC/PHCC 8 (15.4)/12 (23.1)/21 (40.4)/11 (21.2)		
		5 (15.2)/4 (12.1)/18 (54.5)/6 (18.2)	3 (15.8)/8 (42.1)/3 (15.8)/5 (26.3)	
Hepatitis HBV/HCV	0 (0)/0 (0)	3 (5.8)/1 (1.9)		0.402
		1 (3.0)/0 (0)	2 (10.5)/1 (5.3)	0.058
LC	1 (10)	1 (1.9)		0.191
		1 (3.0)	0 (0)	0.171
Hypertension	3 (30)	17 (32.7)		0.870
		14 (42.4)	3 (15.8)	0.245
Diabetes	1 (10)	10 (19.2)		0.492
		8 (24.2)	2 (10.5)	0.771
Smoking none/current/ex-	9 (90)/1 (10)/0 (0)	40 (76.9)/8 (15.4)/4 (7.7)		0.301
		29 (87.9)/3 (33.3)/1 (3.0)	11 (57.9)/5 (26.3)/3 (15.8)	0.012 *
Alcoholic	5 (50)	14 (26.9)		0.152
		8 (24.2)	6 (31.6)	0.477
Dyslipidemia	0 (0)	5 (9.6)		0.314
		3 (9.1)	2 (10.5)	0.382
BMI	23.8 ± 1.7	22.8 ± 3.0		0.283
		23.4 ± 3.0	21.6 ± 2.8	
Laboratory Findings				
WBC	6068.0 ± 1797.6	7492.1 ± 3739.5		0.246
		7504.2 ± 4093.8	7471.1 ± 3134.6	0.512
NLR	2.59 ± 1.86	4.06 ± 4.27		0.290
		4.26 ± 5.12	3.73 ± 2.22	0.519
Hb	12.7 ± 1.2	12.4 ± 1.7		0.718
		12.6± 1.74	12.1 ± 1.74	0.536
PLT (k)	221.6 ± 47.6	270.3 ± 85.3		0.086
		276.3 ± 82.5	259.9 ± 91.4	0.181
ALT	28.6 ± 21.0	119.9 ± 125.0		0.026 *
		140.5 ± 132.5	84.0 ± 104.5	0.020 *
ALP	139.9 ± 163.9	326.7 ± 263.1		0.035 *
		367.9 ± 293.7	255.1 ± 184.9	0.032 *
Total Bilirubin	0.81 ± 3.44	5.76 ± 7.70		0.048 *
		5.60 ± 7.40	6.04 ± 8.38	0.140
Albumin	4.29 ± 0.43	4.01 ± 0.54		0.128
		4.10 ± 0.45	3.84 ± 0.65	0.075
PNI	50.9 ± 5.4	47.6 ± 6.5		0.134
		48.6 ± 5.6	46.0 ± 7.8	0.118
BUN	11.6 ± 3.6	14.4 ± 5.2		0.108
		15.2 ± 4.9	13.0 ± 5.5	0.092
Creatinine	0.72 ± 0.10	0.79 ± 0.21		0.325
		0.83 ± 0.20	0.72 ± 0.22	0.105
C-related protein	1.52 ± 1.57	2.54 ± 3.86		0.421
		2.16 ± 3.65	3.19 ± 4.22	0.446
CEA	3.0 ± 1.2	6.5 ± 9.8		0.430
		3.9 ± 2.8	11.1 ± 15.0	0.021 *
CA19-9	16.0 ± 9.5	701.3 ± 1240.2		0.185
		434.3 ± 930.8	1165.1 ± 1568.3	0.040 *

IHD (intrahepatic duct), CBD (common bile duct), GB (gallbladder), IHCC (intrahepatic cholangiocarcinoma), EHCC (extrahepatic cholangiocarcinoma), HBV (hepatitis B virus), HCV (hepatitis C virus), LC (liver cirrhosis), NLR (neutrophil/lymphocyte ratio), PNI (prognostic nutrition index) *: *p*-value < 0.05.

**Table 2 jcm-10-04435-t002:** CTC count and sensitivity and specificity of each parameter between benign and biliary tract cancer.

	Benign Biliary Disease *n* = 10	Biliary Tract Cancer *n* = 52		*p*-Value
		Resectable *n* = 33	Unresectable *n* = 19	
CTC count	30.9 ± 16.7	125.7 ± 259.8		0.256
		145.8 ± 320.2	90.7 ± 82.9	0.386
CTC count > 40	2 (20)	29 (55.8)		0.039 *
		18 (54.5)	11 (57.9)	0.090
V-CTC	8.6 ± 7.3	39.8 ± 89.6		0.278
		45.6 ± 110.3	29.8 ± 30.4	0.449
V-CTC > 15	1 (10)	30 (57.7)		0.005 *
		17 (51.5)	13 (68.4)	0.004 *
VCR (%)	23.8 ± 11.8	35.7 ± 17.9		0.048 *
		36.2 ± 17.7	34.9 ± 18.6	0.139
VCR > 40% (%)	1 (10)	25 (48.1)		0.025 *
		15 (45.5)	10 (52.6)	0.045 *
Over two of three parameters (1)	1 (10)	32 (61.5)		0.002 *
		19 (68.4)	13 (68.4)	0.005 *
(1) and/or biopsy	1 (10)	47 (90.4)		<0.001 *
		29 (87.9)	18 (94.7)	<0.001 *
(1) and/or CA19-9	1 (10)	47 (90.4)		<0.001 *
		28 (84.8)	19 (100)	<0.001 *
	AUC	Sensitivity		Specificity
Over two of three parameters (1)	0.758	61.5%		90%
Biopsy	0.885	78%		100%
CA19-9 > UNL	0.846	60.6%		100%
(1) and/or biopsy (+)	0.902	90.4%		90%
(1) and/or CA19-9 > UNL	0.902	90.4%		90%

BTC, biliary tract cancer; V-CTC, Vimentin + CTC; VCR, vimentin/CTC ratio; Three parameter (CTC count > 40, V-CTC > 15, VCR > 40%), Biopsy (+): malignancy was proven by biopsy, UNL: upper normal limit, AUC: area under curve, *: *p*-value < 0.05.

**Table 3 jcm-10-04435-t003:** CTC count and sensitivity and specificity of each parameter between benign and resectable biliary tract cancer.

	Benign Biliary Disease *n* = 10	Resectable BTC *n* = 33	*p*-Value
CTC count	30.9 ± 16.7	145.8 ± 320.1	0.194
CTC count > 40	2 (20)	18 (54.5)	0.002 *
V-CTC	8.6 ± 7.3	45.6 ± 110.3	0.087
V-CTC >15	1 (10)	17 (51.5)	<0.001 *
VCR	23.8 ± 11.8	36.18 ± 17.7	0.031 *
VCR > 40%	1 (10)	15 (45.5)	<0.001 *
Over two of three parameter (1)	1 (10)	19 (57.6)	0.007 *
(1) and/or biopsy	1 (10)	29 (87.9)	<0.001 *
(1) and/or CA19-9	1 (10)	28 (84.8)	<0.001 *
	AUC	Sensitivity	Specificity
Over two of three parameter (1)	0.738	57.6%	90%
(1) and/or biopsy	0.889	87.9%	90%
(1) and/or CA19-9	0.874	84.8%	90%

BTC, biliary tract cancer; V-CTC, Vimentin + CTC; VCR, vimentin/CTC ratio; Three parameter (CTC count > 40, V-CTC > 15, VCR > 40%) AUC: area under curve, *: *p*-value < 0.05.

**Table 4 jcm-10-04435-t004:** Prognostic factor analysis via Cox regression analysis.

	Univariable Analysis		Multivariable Analysis	
	HR (95%CI)	*p*-Value	HR (95%CI)	*p*-Value
V-CTC > 50	2.042 (1.006–4.146)	0.048 *	2.172 (1.064–4.433)	0.033 *
CTC count > 40	1.665 (0.927–2.989)	0.088	1.427 (0.717–2.841)	0.311
VCR > 40%	1.154 (0.660–2.016)	0.615	1.030 (0.583–1.820)	0.919
CA19-9 > UNL	1.622 (0.881–2.988)	0.121	1.705 (0.924–3.148)	0.088
NLR > 3.5	1.149 (0.637–2.073)	0.645	1.716 (0.885–3.327)	0.110

HR, Hazard ratio; CI, confidence interval; CTC, circulating tumor cells; V-CTC, Vimentin + CTC; VCR, vimentin/CTC rate; UNL, upper normal limit; NLR, neutrophil-lymphocyte ratio, *: *p*-value < 0.05.

**Table 5 jcm-10-04435-t005:** Baseline characteristics according to V-CTC level.

	V-CTC Over 50 (*n* = 11)	V-CTC Under 50 (*n* = 41)	*p*-Value
Sex male, (%)	5 (45.5)	27 (65.9)	0.225
Age	73.1 ± 12.2	68.1 ± 10.3	0.179
Diagnosis GB/IHCC/EHCC/PHCC	1 (9.1)/3 (27.3)/5 (45.5)/2 (18.2)	7 (17.1)/9 (22.0)/16 (39.0)/9 (22.0)	0.839
hepatitis HBV/HCV	0 (0)/1 (9.1)	3 (7.3)/0 (0)	0.376
Liver cirrhosis	0 (0)	1 (2.4)	0.609
Hypertension	6 (54.5)	11 (26.8)	0.085
Diabetes	3 (27.3)	7 (17.1)	0.456
smoking none/current/ex-	8 (72.7)/3 (27.3)/0 (0)	32 (78.0)/5 (12.2)/4 (9.8)	0.833
alcoholic	3 (27.3)	11 (26.8)	0.977
Dyslipidemia	2 (18.2)	3 (7.3)	0.287
BMI	22.4 ± 2.6	22.9 ± 3.2	0.617
Pathology well-/moder-/poor	0 (0)/2 (18.2)/2 (18.2)	4 (9.8)/17 (41.5)/7 (17.1)	0.031 *
Metastatic	3 (27.3)	9 (22.0)	0.716
Operable	7 (63.6)	26 (63.4)	0.989
Palliative Chemotherapy	2 (18.2)	11 (26.8)	0.565
Op and no recurrence	3 (27.3)	16 (39.0)	0.320
Laboratory Findings			
WBC	6503.6 ±1789.0	7757.3 ± 4085.1	0.328
NLR	3.2 ± 1.3	4.3 ± 4.8	0.444
Hb	11.9 ± 0.9	12.6 ± 1.9	0.292
PLT (k)	241.6 ± 91.2	278.0 ± 83.1	0.213
ALT	124.5 ± 117.8	118.6 ± 128.3	0.890
ALP	349.8 ± 328.0	320.5 ± 247.3	0.746
Total bilirubin	7.81 ± 8.97	5.21 ±7.34	0.323
Albumin	3.93 ± 0.61	4.03 ± 0.53	0.609
PNI	46.7 ± 7.8	47.9 ± 6.2	0.618
BUN	16.7 ± 5.8	13.8 ± 4.9	0.095
Creatinine	0.86 ± 0.23	0.77 ± 0.21	0.250
C-related protein	2.0 ± 2.1	2.7 ± 4.2	0.632
CEA	11.1 ± 18.9	5.2 ± 4.6	0.082
CA19-9	654.5 ± 1198.6	713.9 ± 1265.3	0.889

GB (gallbladder), IHCC (intrahepatic cholangiocarcinoma), EHCC (extrahepatic cholangiocarcinoma), PHCC (perihilar cholangiocarcinoma), HBV (hepatitis B virus), HCV (hepatitis C virus), LC (liver cirrhosis), NLR (neutrophil/lymphocyte ratio), PNI (prognostic nutrition index), *: *p*-value < 0.05.

## Data Availability

All relevant data contained within the article and Appendix A.

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
