# Peer review of "Vimentin-Positive Circulating Tumor Cells as Diagnostic and Prognostic Biomarkers in Patients with Biliary Tract Cancer"

_jcm, 2021, doi:10.3390/jcm10194435_

Round 1
Reviewer 1 Report
This is a nice study on an important and very interesting topic of broad interest. The methodology is sound and the conclusions are justified. However, the manuscript needs extensive editing in terms of language and, to a much lesser extent, presentation of data. A thorough revision of the text will benefit the manuscript significantly!
Major points
1. The abstract needs extensive editing.
Methods
2. CTC counting criteria (line 101 and Figure 2): I understand that CTCs should have an EpCAM+/CK+/CD45- profile and a DAPI+ nucleus.
a)Does intact morphology refer to the shape of the cell or the nucleus? What about the size of the cell- is there a cut-off size for CTCs?
b)It has been reported that not all CTCs express CK-EpCAM because of EMT. What percentage of V-CTCs coexpress Vimentin and CK-EpCAM?
3. Figure 2: You may move this figure to the results section. A legend would be beneficial. The top panel refers to cancer cell line+BC but there is no description of any cell line used in the methods. Is this a cell line for validation experiments?
S045 patient: All depicted DAPI cells are considered as CTC cells? It would be nice to include a similar panel from BBD patient and/or a panel from a patient with CTCs but not V-CTCs for comparison-validation of staining.
Results:
4.What percentage of BBD and BTC patients had CTCs and V-CTCs? How many patients in each group had no CTCs/V-CTCs? This can be illustrated in a graph.
5.Lines 143-144 and Table 2-Figure 3: You mention in the text that there is a significant difference between groups in CTC and V-CTC counts but p-values in the table are >0.05 and there is no indication of significance in the figure. Please clarify.
6. Survival curves: Y-axis is not labeled. Do you show data of OS or PFS? This should be clarified in the text as well.
Discussion:
7. The count of CTCs and V-CTCs is considerably higher than the count reported in other studies. Is it an advantage of the particular platform that you used in comparison with established platforms like CellSearch or other approaches (glycosaminoglycan probes)? How do you validate that these are real tumor cells and not non-specific non-tumor cells?
8.You mention that EpCAM and panCK markers may underestimate CTC count in EMT and vimentin-based methods can increase this yield. This is true for pancreatic cancer as demonstrated by Wei et al https://doi.org/10.1016/j.canlet.2019.C03.009.Your data suggest that the CTC count is higher than V-CTC count in all groups (no statistical analysis provided for this comparison). How do you explain this?
Minor points
Figure 1: The flow chart can be more informative. Basic information should be included in the chart (number of BBD patients, number of BTC patients with resectable and unresectable cancer). Moreover, it should be mentioned that all patients excluded from the study for technical reasons were from the BTC group.
- I think it is better to use p<0.001 rather than p=0.000 in the tables.
Line 47 Why since 2007?
Line 58 Please expand on the possible advantages of using V-CTC as a marker instead of conventional CTC.
Lines 60-61 References are missing.
Lines 90-91 Mention the conjugated antibodies FITC anti-EpCAM, TRITC anti-Vimentin used in the study and provide dilutions.
Line 104 V-CTCs do not determine EMT expression, please rephrase.
Line 110 Please define PFS and OS.
Line 125 Define ROC and mention the statistical package used for the analysis.
Line 138 CEA is missing.
Tables 1+S1 Some abbreviations are missing.
Table 2 heading Replace Bilairy with Biliary
Line 187 Replace Kaphlan with Kaplan
Line 198 V-CTC proportion or V-CTC count?
Line 240 Replace EPCP with ERCP
Reviewer 2 Report
In their study, Han et al. evaluated in a prospectively collected cohort the diagnostic and prognostic role of CTCs detection (and in particular of V-CTC) in a group of patients with biliary tract cancers. They found that CTC count, V-CTC count and V-CTC/total CTC ratio (VCR) are higher in patients with biliary tumors compared to a control group including patients with benign biliary diseases. This was confirmed also restricting the analysis to patients with resectable BTC. Moreover, V-CTC count with a cut-off of 50/mL proved to be able to discriminate patients according their overall survival. Sensitivity and specificity of these markers are probably too low in the diagnostic setting, but they might be useful in prognosis prediction.
Considering that a well-performing biomarker is needed in BTC, this study provides some interesting results. However, I have several concerns.
1) Methods
- I have some concerns on study and control group. Most cholangiocarcinomas arise the novo, and no risk factors are identified. Among the well-established risk factors there are, especially for intrahepatic disease, cirrhosis and viral hepatitis B and C. In addition, cholangiocarcinoma is strongly associated with primary sclerosing cholangitis (PSC). With this premise, I would have included in the study also a group of healthy subjects as controls. In addition, I would have included in the control group also cirrhotics (HBV and HCV) and patients with PSC. How many patients in the BBD and BTC groups have PSC?
- Intrahepatic, perihilar and distal cholangiocarcinomas have distinct epidemiology, biology and prognosis. The Authors included in their study also gallbladder cancers. As a consequence, the BTC group of this study is highly heterogeneous and this probably prevent the possibility to obtain clinically useful results. A sub-analysis according the type of cancer should be provided (at least for the prognostic role of biomarkers), even if the number of patients in each subgroup is probably too low to obtain meaningful results.
- In the “Statistical analysis” paragraph is not clear which tests have been used for comparison between groups. In the comparison of categorical variables chi-square test and Fischer’s exact test must be used. When quantitative data are compared, Student’s t test and ANOVA test are appropriate for normally distributed variables (comparison between two groups or three or more groups, respectively). Non-parametric Wilcoxon rank sum test and Kruskal-Wallis test are used in the comparison of 2 or ≥3 or more groups when variables are not-normally distributed. Please clarify this in the text.
- The Authors excluded from the analysis 7 patients in whom no CTCs were detected. This is appropriate if there were some technical problems preventing the quantification of CTCs. On the contrary, if no CTC were detected following an experiment without technical issues, these patients should be included in the analysis. In fact, the absence of CTCs in some patients is a result that can influence sensitivity and specificity.
2) Results:
- In lines 143-144, the Authors state that CTC count and V-CTC count were significantly higher in BTC patients than in patients with BBDs (p<0.001). However, this data is in contrast with what is reported in Table 2, where no differences between the two groups are shown. What are the correct results?
- Diagnostic cut-offs of CTCs, V-CTCs and VCR have been established with ROC curve method. I suppose that these values were those maximizing sensitivity and specificity. ROC curves should be reported in the paper. Moreover, sensitivity, specificity, predictive values and AUC of the determination of CTCs, V-CTCs and VCR should be clearly reported in the results.
- I understand the rationale behind testing the diagnostic role of the combination between biomarkers. However, this is not the case for the combination of circulating biomarkers with biopsy results. First, it is not clear what the Authors means with this analysis. If all patients included in the study had a histological confirmation of their cancer, the sensitivity of ≥2 markers positive and the sensitivity of ≥2 markers positive + positive biopsy should be the same. Second, what should be the clinical significance or implication of this combined evaluation? The evaluation of circulating biomarkers provides no advantage as compared to histological sampling in this study (from which this conclusion cannot be drawn as all patients included have their diagnosis confirmed with biopsy).
- Were all the BTC patients included in the study diagnosed histologically (as stated in methods)? If this is the case, the sensitivity of biopsy should be 100% by definition (and not 78% as reported in Table 2). Please clarify. Also the sensitivity reported in this table for the combination of ≥2 parameters (57.7%) is different from what reported few lines above (61.5%). Which is the correct result?
- If all patients included had a histological diagnosis, I cannot understand the results reported in the row Pathology of Table 4 (the sum should be 52).
- In the survival analysis, also the prognostic role of CTC and VCR should be evaluated and, if no significant differences are found, this should be stated in a brief sentence. Have you considered to perform a survival analysis using the combination of CTC, V-CTC and VCR? Moreover, the performance of these biomarkers should be compared to the prognostic performance of CA 19.9.
- How was determined the cut-off of 50/mL used for V-CTC? What does it mean that this value was the most meaningful? Please specify.
- Please indicate confidence intervals of median survival.
- The evaluation of the V-CTC role as predictive of PFS is mentioned as secondary endpoint in methods. However, I cannot find any result regarding PFS.
- V-CTC value is associated with prognosis of BTC patients at univariate analysis. To evaluate if this is a strong and independent predictor of survival, I suggest to perform a multivariable Cox regression analysis including other predictors of survival (for instance treatment).
3) Discussion is not easy to follow and need an extensive revision. The Authors should discuss their results, in particular acknowledging that evaluation of CTCs, V-CTCs and VCR has low accuracy in diagnosing BTC. Probably these markers are more useful in the prognostic setting, but additional analysis are needed to support this conclusion.
4) Tables: please define the abbreviations used in the tables.
Round 2
Reviewer 1 Report
The authors addressed my questions and concerns satisfactorily and the revised version is much improved in terms of content and clarity of presentation.
In my opinion there are two issues regarding data presentation that need to be addressed before publication.
- Figure 3:
I think that the figure is mislabeled. In both versions (original and revised) the groups are labeled as resectable n=19 and unresectable n=33 but this is in contrast with numbers provided elsewhere. Moreover, the group labeled as resectable in the original version is labeled as unresectable in the revised and vice versa. Based on Table 2+ Figure 1+methods section, I think that the current-revised figure is the right one but the mid group should be labeled unresectable (n=19) and the right group should be labeled resectable (n=33).
- Supplementary table 2 is confusing and needs editing. It is not clear which group of patients (n=35) is analyzed
Minor points
- Please update the supplementary materials description (line 423)
- A minor editing/spell check is needed. For example line 99 1:417 dilution is a typo?, line 118 better use intact instead of clear etc. The abstract is now well written but the lines 26-29 can be rephrased to avoid repetition.
Author Response
The authors addressed my questions and concerns satisfactorily and the revised version is much improved in terms of content and clarity of presentation.
In my opinion there are two issues regarding data presentation that need to be addressed before publication.
- Figure 3:
I think that the figure is mislabeled. In both versions (original and revised) the groups are labeled as resectable n=19 and unresectable n=33 but this is in contrast with numbers provided elsewhere. Moreover, the group labeled as resectable in the original version is labeled as unresectable in the revised and vice versa. Based on Table 2+ Figure 1+methods section, I think that the current-revised figure is the right one but the mid group should be labeled unresectable (n=19) and the right group should be labeled resectable (n=33).
Reply: Thank you for your comment. You’re right. We edited it in figure 3.
- Supplementary table 2 is confusing and needs editing. It is not clear which group of patients (n=35) is analyzed
Reply: Thank you for your comment. The 35 patients underwent curative surgery or palliative chemotherapy. We edited supplementary table 2 description as below.
“comparison of PFS according to CTC marker and CA19-9 in 35 patients who underwent curative surgery or palliative chemotherapy”
Minor points
- Please update the supplementary materials description (line 423)
Reply: Thank you for your comment. We added supplementary materials description.
- A minor editing/spell check is needed. For example line 99 1:417 dilution is a typo?, line 118 better use intact instead of clear etc. The abstract is now well written but the lines 26-29 can be rephrased to avoid repetition.
Reply: Thank you for your comment. We made a mater mix solution of the antibodies. Therefore, the 1:417 dilution ratio is correct. We edited it as your comments and rephased the abstract as follow:
“V-CTC could be a potential biomarker for early diagnosis and predicting prognosis in patients with BTC.”
Reviewer 2 Report
The Authors answered adequately to all my questions and the overall quality of manuscript, my opinion, has significantly improved.
I have only one other comment: in Table 4, the Authors included OR as the association measure of the Cox multivariate analysis. However, the correct association measure is hazard ratio (HR). In addition, results of univariate analysis should be showed in the Table. These adjunctive analysis should be described in Methods.
Author Response
The Authors answered adequately to all my questions and the overall quality of manuscript, my opinion, has significantly improved.
I have only one other comment: in Table 4, the Authors included OR as the association measure of the Cox multivariate analysis. However, the correct association measure is hazard ratio (HR). In addition, results of univariate analysis should be showed in the Table. These adjunctive analysis should be described in Methods.
Reply: Thank you for your comment. We edited table 4. as your comment, and added below sentence in statistical analysis at method
“To evaluate the factors affecting the prognosis, COX regression analysis was performed including factors known as prognostic markers and CTC markers as variables.”.